# Effects of Butterfly Pea Extracts on Phagocytic Activity of Blood Polymorphonuclear Leukocytes and Muscular Lipid Peroxidation in Rabbits

**DOI:** 10.3390/ani14060958

**Published:** 2024-03-20

**Authors:** Attawit Kovitvadhi, Laura Gasco, Ivo Zoccarato, Theera Rukkwamsuk

**Affiliations:** 1Department of Agriculture, Forest, and Food Sciences, University of Turin, Largo P. Braccini 2, 10095 Turin, Italy; fvetawk@ku.ac.th (A.K.); laura.gasco@unito.it (L.G.); ivo.zoccarato@unito.it (I.Z.); 2Department of Physiology, Faculty of Veterinary Medicine, Kasetsart University, Bangkok 10900, Thailand; 3Institute of Science of Food Production, National Research Council, Largo P. Braccini 2, 10095 Turin, Italy; 4Department of Large Animal and Wildlife Clinical Sciences, Faculty of Veterinary Medicine, Kasetsart University, Nakhon Pathom 73140, Thailand

**Keywords:** antioxidant, *Clitoria ternatea*, digestibility, lipid peroxidation, phagocytosis

## Abstract

**Simple Summary:**

Tropical herb extracts, which could be used to provide health benefits, might be an effective strategy to promote benefits for rabbit farming. The present study indicated that the supplementation of rabbit diets with butterfly pea (*Clitoria ternatea* L.) crude extract had no effect on phagocytic function but did improve digestibility. Modification of the meat color and delayed lipid peroxidation were observed. Supplementation with *Clitoria ternatea* L. crude extract at 0.5 g/kg body weight could be useful in the production of rabbits for meat consumption, prolonging shelf life by delaying lipid peroxidation.

**Abstract:**

Sixteen 35-day-old male crossbred rabbits (New Zealand white × Thai native breed) with an initial weight of 484 ± 11.3 g were randomly divided into two groups of eight, constituting control and treatment groups. The treatment group was orally administered a crude extract of butterfly pea (*Clitoria ternatea* L.) at 0.5 g/kg body weight from weaning (at 35 days) to slaughter (at 90 days). The effects on the phagocytic activity of blood polymorphonuclear leukocytes, serum biochemistry, meat quality, muscular lipid peroxidation, the apparent digestibility of dry matter and nutrients, and gut histology were studied. The results revealed that the phagocytic function of circulating leukocytes (75 and 90 days) and alveolar macrophages (90 days) did not differ between the two groups. At slaughter, treated rabbits had lower blood urea nitrogen concentrations and higher liver weight than control rabbits (*p* < 0.05). After chilling at 4 °C for 24 h, a lower meat pH and the alteration of meat color (brighter, less yellow, lower hue angle, and decreased color saturation) were observed in the treated group (*p* < 0.05). Furthermore, lipid peroxidation (measured at 3, 5, and 7 storage days) in the meat of treated rabbits was lower than in controls (*p* < 0.05). The apparent digestibility of organic matter and ether extract (analyzed at 46 days for 4 days) was improved in the treated group (*p* < 0.05), whereas gut histology was unaffected. In conclusion, butterfly pea extract supplementation did not affect phagocytic function but led to a modification in meat color, delayed lipid peroxidation, and improved digestibility.

## 1. Introduction

An excess of reactive oxygen species and/or nitrogen species beyond the cellular antioxidant capacity can result in oxidative damage to lipids, proteins, and DNA, leading to cell membrane damage, the loss of enzyme activities, and mutagenesis [1]. Free radical molecules adversely affect health status and cause many major diseases [1], including digestive diseases [2]. Oxidative stress is essential in the pathogenesis and progression of inflammatory bowel disease and its relationship with the mucosal immune response [3]. Consequently, oxidative stress generates inflammation in the intestinal wall and causes the flattening of the gross folds of the intestinal lining, leading to a loss of surface area and a syndrome marked by a reduced ability to digest and absorb nutrients.

In rabbits, the causes of this syndrome are often multifactorial. An inappropriate diet or environmental conditions put them at greater risk of developing it [4]. The excessive production of reactive oxygen species that cannot be counteracted by the action of antioxidant enzymes in the body causes many diseases; the imbalance between oxidant species and the antioxidant defense system can trigger specific factors that are responsible for oxidative damage in the cell [5]. Therefore, antioxidant substances have been used to prevent or reduce damage to the gastrointestinal tract caused by oxidative stress. The resulting benefits of longevity, immunity, and health should interest those involved in the pet and fur trades because they keep rabbits for a long time. Furthermore, supplementing feed with some antioxidants could improve productive performance, health, and meat quality, which should interest rabbit meat producers [6].

Synthetic antioxidants are commonly used in the food industry to prevent the deterioration of meat products. However, their use is increasingly limited because they are considered highly unstable. Furthermore, relevant legislation is becoming very strict, as they are suspected to have carcinogenetic and mutagenic properties [7]. Recently, there has been increasing interest in natural antioxidants extracted from plants, which can be used as alternatives to synthetic antioxidants; consumers prefer these natural and organic additives and ingredients because they are perceived as healthier and more sustainable [8,9,10,11]. Phenolic compounds are secondary metabolites in plants with significant antioxidant and chelating properties; they have been widely studied in the production of livestock and rabbits [8,9,10,11]. Therefore, it could be hypothesized that plant-based dietary supplementation could be an alternative strategy to provide positive results without side effects.

*Clitoria ternatea* (CT) is an ornamental perennial climber from the Fabaceae family, commonly known as butterfly pea. It is native to tropical equatorial Asia, including Thailand. In human medicine, CT is thought to have been used in Ayurvedic medicine, one of the world’s oldest medical systems [8,9,10,11]. A range of properties—including antioxidant, anti-inflammatory, analgesic, antipyretic, and antimicrobial properties—have also been reported in CT studies [8,9,10,11], probably due to the presence of flavanol and anthocyanin glycosides, triterpenoids, tannins, rutin, and phytosterols [8,9,10,11]. Supplementation with CTE has been studied in broilers [12] and layers [13] but not in rabbits.

In rabbits, CT has been used as a protein source rather than for its antioxidant abilities and has been offered directly to rabbits rather than as an extract [14,15]. Rabbit production in Thailand is intensive (50–200 reproductive uses), but it is a primary job for most farmers. However, health problems—mainly digestive and respiratory disorders—have been observed, with negative consequences [16]. From this point of view, using tropical herb extracts to provide health benefits might be an effective strategy. Therefore, this study evaluated the possible effects of CT crude extract (CTE) supplementation on rabbits’ immunity and health status. Phagocytic activity, serum biochemistry, and intestinal histology were investigated. Furthermore, considering the importance of rabbits in human nutrition, digestibility and meat quality were also evaluated.

## 2. Materials and Methods

The experiment was approved by the Institutional Animal Care and Use Committee of Kasetsart University, Bangkok, Thailand (ACKU61-VET-001).

### 2.1. Plant Preparation and Extraction

CT (purple pedal breed) was planted at the Chiang Mai Animal Husbandry and Research Center, Chiang Mai, Thailand. Plants were harvested at 90 days (during the flowering period). The extraction method was described by Suwannamanee [13]. Briefly, whole plants (including leaves and flowers, but not roots) were chopped, mixed, milled using rubber-edged mills with water for 30 min, and filtered. Subsequently, the solid part was separated from the liquid part by heating at 60 °C for 40 min. The solid part was dried by incubating at 60 °C for 72 h and then powdered and kept in a desiccator at 4 °C until use.

### 2.2. Animals, Housing, Diet, and Recording

The experiment was carried out in the experimental facility of the Department of Large Animal and Wildlife Clinical Sciences, Faculty of Veterinary Medicine, Kasetsart University, Nakhon Pathom, Thailand. Sixteen 35-day-old male crossbred rabbits (New Zealand white × Thai native breed; 484 ± 11.3 g) were randomly housed in individual wire cages (L × W × H: 45 cm × 45 cm × 45 cm) inside the facility and maintained at 23 ± 2 °C (12 h light/12 h dark) for the experimental period of 55 days. The rabbits were separated into two groups of eight individuals. CTE in water (0.5 g/kg body weight) was administered orally to the treated rabbits once a day at 09:00 h for the entire experimental period; water (placebo) was administered to the control rabbits. A commercial pellet diet (Lee Feed Mill, Publ. Co., Ltd., Phetchaburi, Thailand) and clean water were provided ad libitum. The diets and CTE were analyzed in duplicate for dry matter (DM), crude protein (CP), ether extract (EE), crude fiber, and ash by ignition, according to guidelines from the Association of Official Analytical Chemists [17]. The neutral detergent fiber, acid detergent fiber, and acid detergent lignin were determined according to the procedures of Van Soest et al. [18]. The level of starch was determined using Ewer’s polarimetric method [19]. The rabbits’ live weight and feed intake were checked once every two weeks to obtain preliminary data on growth performance. Mortality and morbidity were controlled daily by the same observer, according to Gidenne et al. [20].

### 2.3. Apparent Digestibility and Digestive Tract Histology

According to the procedure of Kovitvadhi et al. [15], feces were collected daily at 46 days for 4 days (*n* = 8 per treatment), weighed, and then stored at −20 °C for chemical analysis in duplicate, recording DM, ash, EE, and CP, according to the A.O.A.C. [17]. The apparent digestibility of DM, organic matter, CP, and EE was calculated using the European standardized method [21]. A 10 mm length of the mid-jejunum and ileum (*n* = 8 per group) was obtained after slaughter at 90 days and preserved in 10% buffered neutral formaldehyde solution (pH 7.2–7.4). Tissue samples were processed, embedded in paraffin, sectioned with an 8 µm thickness using a rotary microtome (Shandon AS325, Southeast Pathology Instrument Services, Charleston, SC, USA), and stained with hematoxylin and eosin staining method. The height of the villi and the depth of the crypts were measured under a BX53 light microscope (Olympus, Tokyo, Japan) using an image analysis program (CellSens standard, Olympus Ver 01 07 imaging software, Tokyo, Japan).

### 2.4. Phagocytosis Assay and Serum Biochemistry

At 75 and 90 days, blood samples were taken from the lateral saphenous vein (eight rabbits per group) and collected in a sterile tube containing heparin, whereas alveolar macrophages were taken by flushing 10 mL of Roswell Park Memorial Institute 1640 Medium (RPMI; Sigma-Aldrich, St Louis, MO, USA) into the lungs of slaughtered rabbits. Both samples were performed in duplicate, kept at 4 °C, and immediately transported to the laboratory for the phagocytosis assay. The polymorphonuclear leukocyte fraction was isolated from whole blood using Polymorphprep^TM^ (Axis-shield, Oslo, Norway). This fraction and the alveolar macrophages were harvested and resuspended in RPMI medium with 10% fetal bovine serum (Sigma-Aldrich, Saint Louis, MO, USA) to reach concentrations of 75 × 10^4^ and 20 × 10^4^ cells/mL, respectively. These mixtures were incubated overnight in a 96-well plate at 37 °C in 5% carbon dioxide. Subsequently, the phagocytosis assay was performed following the manufacturer’s instructions (CytoSelect TM, Cell Biolabs Inc., San Diego, CA, USA), and the absorbance was read by a spectrophotometer (Infinite^®^ F50, Tecan, Männedorf, Switzerland) at 450 nm. The results are expressed as optical density from ingested bacterial cells. Serum biochemistry was analyzed using standard protocols (Kasetsart University Veterinary Teaching Hospital, Kasetsart University, Nakorn Pathom, Thailand), where blood samples were collected from eight rabbits per group during slaughter.

### 2.5. Meat Quality and Lipid Peroxidation

The rabbits were euthanized by electric stunning, followed by a jugular vein incision. The carcasses were prepared following the indications of Kovitvadhi et al. [15]. After 24 h of chilling, eight carcasses per group were halved, and then two longissimus dorsi (LD) muscles were excised. The left part of the LD muscle was used to assess meat pH after 24 h of chilling (pH_24_). An ultra-basic portable pH meter (Denver Instrument^TM^, Bohemia, NY, USA) was used, with a homogenized mixture of 10 g of LD and 100 mL of distilled deionized water to establish color (Color checker NR-1, Nippon Denshoku Industries, Tokyo, Japan). The approximate chemical composition (moisture, CP, EE, and ash) was then analyzed according to the procedure of Kovitvadhi et al. [15]. The right part of the LD was separated into four pieces that were then covered with polyvinyl chloride wrap to perform the thiobarbituric acid-reactive substances (TBARS) assay after preservation at 4 °C for 1, 3, 5, or 7 days. Fresh meat (5 g, *n* = 8 per group, in duplicate) was homogenized in 25 mL of 7.5% trichloroacetic acid using a Polytron tissue homogenizer (Type PT 10-35, Kinematica GmbH, Luzern, Switzerland). Lipid peroxidation was determined using a modified TBARS method according to the protocol of Witte et al. [22]. The results are expressed as µg of malondialdehyde per kg of fresh meat, using a standard curve that covered a concentration range of 0.5–10 µM 1,1,3,3-tetramethoxypropane (Sigma-Aldrich, Steinheim, Germany). Absorbance was measured at 532 nm using a UV-1601 spectrophotometer (Shimadzu Corporation, Kyoto, Japan).

### 2.6. Statistical Analysis

All statistical analyses were performed using the R-studio software version 2023.12.1+402 with the package Rcmdr. Statistically significant differences between the groups in each parameter were analyzed using the Student *t*-test, while Welch’s *t*-test was applied when groups had unequal variances. Statistical significance was established at *p* < 0.05.

## 3. Results and Discussion

### 3.1. The Chemical Composition of the Diets and CTE

The chemical composition of the diets and CTE was analyzed and is presented in Table 1. The CTE contained a high level of crude protein, followed by ether extract and ash. Supplementation was at a very low level, around 1% of the diet if we calculate it for a 2 kg rabbit with a feed intake of 100 g/day. Therefore, these nutrients could not interfere with the major nutrients. The active ingredients were not determined in this study; therefore, the consequences could be explained by the CTE. The CP, ash, EE, and crude fiber contents in butterfly pea without extraction have been reported to be in the ranges of 25.11–28.62%, 6.71–8.36%, 3.33–3.61%, and 15.78–19.66%, respectively [23]. The extraction process can lead to an increase in the leaf protein concentrate (LPC) in the CTE used in this study. Many other plant leaves, like cowpea, berry, sweet potato, neem leaves, etc., are useful for the extraction of leaf proteins. The LPC could become a very useful alternative source of protein for animal production [24]. LPCs from different biomass sources had a protein content ranging from 67.2 to 87.7% [25]. However, in the current study, the chemical composition of the CTE was similar to that in previous studies of butterfly pea extract [13]. In general, the highest protein content in the LPC depends on the processing of the extraction with the chemical solution and the purity of the LPC, as other phytochemical compounds, e.g., polyphenolic compounds, may strongly associate with the leaf proteins [25], as well as other chemical components of the primary metabolites of plants.

### 3.2. Apparent Digestibility and Digestive Tract Histology

No statistically significant differences were observed in terms of performance indicators. Slaughter weights at 90 days between the control and CTE groups were 2279 and 2275 g (*p* = 0.98). Additionally, comparable results between the control and CTE groups were found for daily weight gain (32.8 vs. 32.6 g/d; *p* = 0.95), daily feed intake (94.7 vs. 106 g/d; *p* = 0.29), and the feed conversion ratio (2.91 vs. 3.28; *p* = 0.13). It is important to note that, due to the limited number of rabbits per group, these findings provide only preliminary data on growth performance. As such, conclusive statements regarding the effect of CTE on rabbit performance could not be made based solely on this information. However, the digestibility of organic matter and EE increased significantly (*p* < 0.05) in rabbits fed CTE compared to the control group (Table 2). Differences in the digestibility of DM and CP showed a slight trend toward significance (*p* = 0.09 and 0.07, respectively). An improvement in the digestibility of EE was observed in rabbits and broilers after supplementation with thyme (*Thymus vulgaris*) [14], essential oils of thyme and star anise as the main active components [26], and an essential oil blend of oregano, cinnamon, and pepper [27]. Digestibility may be improved by active compounds and/or functions that these plants share with CT. However, a study on the mechanisms of each chemical compound should be performed. Gut histology (villi height and crypt depth) is another possible indicator of digestive function, but there were no statistically significant differences in this study (Table 2). However, the villi height and crypt depth of the jejunum and ilium of CTE-fed rabbits tended to increase compared to control rabbits, possibly indicating improved digestion and absorption in the intestinal tract. Intestinal crypts are invaginations of the epithelium around the villi. They are lined by enzyme-secreting epithelial cells, and the bases of the crypts are constantly dividing to maintain the villi structure [28]. No morbidity or mortality was observed during the experimental period, reflecting good environmental conditions. The current study indicated that the CTE-containing supplement may be safe and support gut health. A healthy gut includes multiple positive aspects of the gastrointestinal tract, specifically effective digestive and absorption functions of the intestinal brush border and villi improvement with phenolic and flavonoids, especially anthocyanin [29], found in flowers (4.72 mg/L) and leaves (2.92 mg/L) [30]. The phenolic content was found in flowers and leaves with values of 37.58 and 56.74 GAE/g, respectively. The leaves had the greatest total flavonoid content (5.69 mg QE/g), while the lowest level of flavonoids was found in the flowers (1.45 mg QE/g) [30]. These secondary metabolites are compounds in specific cells that are indirectly essential for respiratory photosynthesis or the metabolism and survival of plants, as they protect plant tissues from injuries and insect and animal attacks. Phenolics and flavonoids are considered abundant classes of phytochemicals with health-promoting qualities and functions [31] that possess antibacterial activity against a wide range of microbial species, including pathogenic bacteria, the main cause of disorders of the digestive tract [32]. Oral secondary metabolite administration to animals improves the intestinal barrier integrity and function, with a reduction in proinflammatory molecules, improvement in the tight-junction protein expression, and improvement in antioxidant intracellular activity [33]. However, some phytochemical compounds in plants can have both adverse and beneficial health effects in animals; these are anti-nutritional factors. Anti-nutritional factors exhibit beneficial attributes at optimal levels. Therefore, higher amounts of anti-nutrients may cause adverse effects on animal and human health [20].

### 3.3. Phagocytic Function and Serum Biochemistry

No statistically significant differences were observed regarding phagocytic activities (Table 2). Phagocytosis is a nonspecific defense mechanism with five stages. Both neutrophils and macrophages have similar mechanisms during the destruction stage. A respiratory burst involves the production of potent oxidants in a chain reaction to eradicate ingested pathogens, working with several enzymes. In a normal situation, glutathione, redox-active metals, and antioxidants such as vitamins C and E have enough potential to detoxify the end products after the destructive mechanism [34]. However, unsuitable conditions can cause the increased formation of oxidative molecules beyond the protective level, damaging cellular biomolecules [6]. Therefore, supplementing the diet with exogenous antioxidants should optimize the eradication mechanism, improving phagocytic function. Unfortunately, phagocytic activity was not improved in either circulating leukocytes or alveolar macrophages after CTE supplementation in this study. The reason may be that the experiment was conducted in normal conditions, as the formation of reactive oxygen species did not exceed the cellular antioxidant capacity. Therefore, clear effects on phagocytic function after supplementation should be observed in experiments with poor conditions, pathogen infection, or stimulated stress, each promoting increased oxidative stress.

In this study, blood chemistry was used to establish the physiological and biochemical status of the rabbits. Alanine aminotransferase (AST) and aspartate aminotransferase (ALT) were present in lower concentrations in treated rabbits than in the control group, but these differences were not significant (50.3 vs. 60.2 U/L, *p* = 0.52, and 35.7 vs. 53.2 U/L, *p* = 0.20, respectively; Table 2). The hepatoprotective property of AST and ALT reduction, induced by acetaminophen with CTE supplementation at 200 mg/kg [35] and other phyto-additions [36], was reported in mice. The unclear result in the present study might be due to the study design, which did not include the induction of hepatotoxicity [36]. A decrease in blood nitrogen and creatinine was observed in rabbits with CTE supplementation compared to the control group (12.0 vs. 14.8%, *p* = 0.05, and 0.86 vs. 1.08%, *p* = 0.12, respectively; Table 2), which was similar to a study on CTE addition to alloxan-induced diabetic rats [37]. These results suggest that CTE may contain hepatoprotective or renal properties, although further study is required to confirm these benefits. Other parameters of serum biochemistry were unaffected by the treatment.

### 3.4. Meat Quality and Lipid Peroxidation

The treated group was found to have heavier livers (6.78 vs. 5.41% of cold carcass weight, SEM = 0.33, *p* = 0.03), while carcass traits and meat chemical composition did not differ between groups (Table 2). The increased liver weight in treated rabbits may be due to enhanced liver function and/or fat storage. However, there were no clinical signs or gross lesions due to hepatic lipidosis or liver enlargement in the treated rabbits. In addition, serum AST and ALT, which indicate liver damage, did not increase. For these reasons, it can be concluded that CTE supplementation did not adversely affect liver function. CTE supplementation decreased pH_24_, yellowness (b*), hue angle (H*), and chroma (C*) while increasing the lightness (L*) of rabbit meat (Table 2). Meat color influences consumers’ purchasing decisions, and surface discoloration can lead to rejection by consumers, directly influencing economic losses. In most studies, antioxidant supplementation did not influence meat color [6]. However, higher meat brightness was reported after the dietary addition of vitamin E in rabbits [38] and of CTE in broilers [13] and the rabbits in this study. The reason may be the protection of the cellular membrane by antioxidants, reducing the loss of liquids into the extracellular space and thus increasing light refraction and maintaining a bright color. Furthermore, this mechanism might lead to higher moisture in the meat of treated rabbits, increasing the meat’s brightness (Table 2). Lo Fiego et al. [38] found a negative correlation between pH_24_ and L* in rabbit meat, as was also found in the present study. The lower pH_24_ in CTE-treated rabbits should be beneficial because proteolytic bacteria developed rapidly when the pH value was around 6 [39]. Lower yellowness in the CTE-treated group may lead to a smaller hue angle and lower saturation. The small amount of fat in the meat of treated rabbits may be related to a lower yellowness value compared to the control group. Rabbit meat is considered healthy from a commercial point of view [39]; therefore, brighter and less yellow meat should attract the attention of consumers who aim to eat healthily.

The level of lipid peroxidation did not differ between the two groups on the first storage day, but this chemical reaction was clearly delayed in the treated group after 3, 5, and 7 storage days compared to the control group (*p* < 0.05; Figure 1). The level of lipid oxidation involves three stages: initiation, propagation, and termination. The first day of storage is the initiation stage of the reaction. Initiation is frequently attributed to the reaction of fatty acids with active oxygen species; in this stage, the accumulation of levels of lipid oxidation products is low, and free radicals preferentially oxidize the natural antioxidants present in meat. This process protects fatty acids in the earliest stage of oxidation or the β-scission reaction, which is minimal during this stage but increases exponentially during the propagation and termination stages [40] after the third day of storage. Antioxidant materials used in meat products are mainly composed of phenolic compounds and flavonoids, which are able to inhibit the lipid peroxidation of meat products, thereby preserving meat quality [41]. Phenolic compounds have strong hydrogen-radical-donating activity, and the presence of the aromatic hydroxyl group is a critical determinant of their H donation and free radical scavenging activity, while free OH flavonoid groups scavenge free radicals and chelate metal ions, including Fe^2+^, Fe^3+^, and Cu^2+^, that bind to chain-formation-initiating catalysts [41]. High levels of polyunsaturated fatty acids are characteristic of rabbit meat, reducing meat quality during storage in terms of nutritive value and safety from lipid peroxidation. However, supplementation with antioxidant-containing substances can prevent or slow this reduction. In an in vitro study, CTE had antioxidant activities in a 2,2-diphenyl-1-picrylhydrazyl radical (DPPH) scavenging assay, and it was suggested that anthocyanin is the main active chemical compound responsible for these activities [35,42,43]. Increased DPPH in broiler thigh and breast meat was reported following dietary CTE supplementation of 0.25 or 0.5% [13], the same level as in the present study. These beneficial effects in preventing lipid peroxidation in meat should confirm the bioavailability of CTE following oral supplementation. However, further studies should be performed to isolate the responsible compounds and determine their modes of action.

There was a limitation in this study. The number of animals per group appears to be low for measuring performance parameters, but it is important to note that the focus of this study was not primarily on performance metrics, as we present these as preliminary results. Instead, our study focused on evaluating phagocytic function, serum biochemistry, digestibility, gut histology, meat color, and lipid peroxidation in meat. With this focus, a sample size of eight rabbits per group was deemed sufficient to discern meaningful outcomes. Significantly different results were observed in organic matter digestibility, ether extract digestibility, meat color, and lipid peroxidation (*p* < 0.05). Hence, the number of animals per group adequately captured the effects of the supplements under investigation. Furthermore, the power of analysis (1-β) exceeded 0.8 for all parameters of interest, further bolstering the robustness of our findings. However, further study on a higher number of animals could be performed.

## 4. Conclusions

This study indicated that supplementation with CT crude extract in rabbit diets had no effect on phagocytic function but did improve the apparent digestibility of organic matter and ether extract. Modification of the meat color and delayed lipid peroxidation were also observed. Therefore, supplementation at 0.5 g/kg body weight could be useful in the production of rabbits for meat consumption, prolonging shelf life by delaying lipid peroxidation.

## Figures and Tables

**Figure 1 animals-14-00958-f001:**
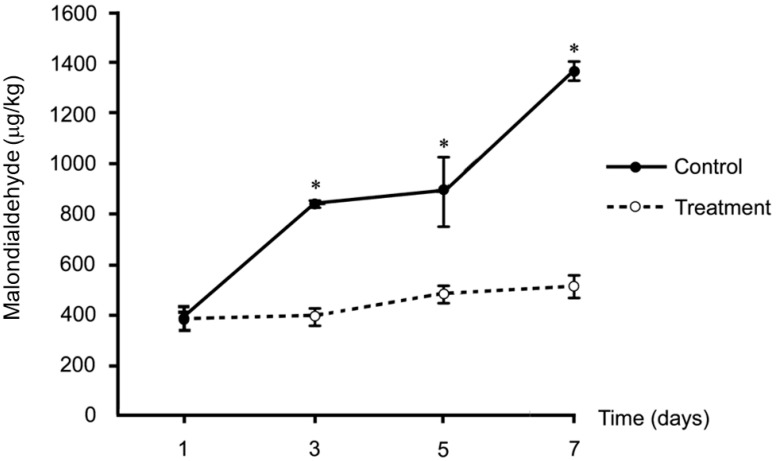
Muscular lipid peroxidation—malondialdehyde (MDA) formation in control (*n* = 8) and *Clitoria ternatea* extract (CTE)-treated (*n* = 8) rabbits during 7 days of storage. Asterisks (*) indicate that the mean MDA between control and treatment (CTE-treated) group at each sampling time were significantly different (*p* < 0.05).

**Table 1 animals-14-00958-t001:** Chemical composition on a dry matter basis (%) of the experimental diets (control) and *Clitoria ternatea* crude extract (CTE).

Chemical Components	Control	CTE
Metabolizable energy (kcal/kg)	2570	-
Dry matter	90.1	91.6
Crude protein (CP)	16.5	43.7
Ether extract (EE)	2.38	10.2
Ash	7.17	11.4
Crude fiber	13.0	2.90
Neutral detergent fiber (NDF)	27.7	-
Acid detergent fiber (ADF)	17.7	-
Acid detergent lignin (ADL)	4.53	-
Starch	29.7	23.4

**Table 2 animals-14-00958-t002:** Effects of *Clitoria ternatea* extract (CTE) on the parameters of apparent digestibility, health status, and some qualities of meat in rabbits.

Parameters	Groups		SEM ^1^	*p*-Value
	Control (*n* = 8)	CTE (*n* = 8)		
Apparent digestibility (%)				
Dry matter	66.0	68.5	0.71	0.09
Organic matter	57.9	67.9	2.62	0.04
Ether extract	55.4	61.7	2.15	0.003
Crude protein	93.4	94.8	0.39	0.07
Leukocyte phagocytic activity ^2^				
75 days	0.87	1.06	0.07	0.17
90 days	1.26	1.42	0.06	0.50
Alveolar macrophage activity ^2^	0.97	0.86	0.06	0.43
Serum biochemistry				
Total protein (g%)	5.78	5.55	0.13	0.42
Albumin (g%)	3.35	3.28	0.05	0.59
Globulin (g%)	2.43	2.27	0.11	0.51
Aspartate aminotransferase (AST, U/L)	60.2	50.3	7.05	0.52
Alanine aminotransferase (ALT, U/L)	53.2	35.7	6.67	0.20
Blood urea nitrogen (g%)	14.8	12.0	0.69	0.05
Creatinine (g%)	1.08	0.86	0.06	0.12
Cholesterol (g%)	96.5	88.5	11.2	0.74
Villus height and crypt depth (µm)				
Jejunal height	474	476	20.0	0.97
Ileal height	623	658	35.3	0.64
Jejunal crypt	108	114	4.27	0.49
Ileal crypt	122	124	3.30	0.74
Carcass characteristics				
Skin, paws, and feet (%SW)	18.0	18.4	0.31	0.49
Full gastrointestinal tract (%SW)	22.3	23.7	1.40	0.63
Cold carcass weight (CCW, g)	1397	1367	60.0	0.82
Dressing percentage (%)	61.2	59.3	0.73	0.23
Liver (%CCW)	5.41	6.78	0.33	0.03
Kidneys (%CCW)	1.14	1.18	0.05	0.76
Perirenal fat (%CCW)	2.35	2.02	0.22	0.51
Thoracic organs (%CCW)	1.49	1.40	0.05	0.69
Cecal trials				
Full caecum (%SW)	6.84	7.45	0.55	0.60
Empty caecum (%SW)	1.52	1.61	0.09	0.65
Cecal content (%SW)	5.31	5.84	0.47	0.60
Cecal pH	6.57	6.45	0.07	0.46
Meat quality				
pH_24_	5.89	5.79	0.01	0.005
Lightness (L*)	52.1	56.2	0.60	0.001
Redness (a*)	1.54	1.60	0.04	0.49
Yellowness (b*)	3.50	1.71	0.36	0.01
Hue (H*)	60.7	33.7	5.09	0.006
Chroma (C*)	3.93	2.81	0.27	0.03
Moisture (%fresh meat)	76.4	77.1	0.23	0.13
Protein (%dry matter)	92.8	92.4	0.23	0.39
Ether extract (%dry matter)	1.86	1.82	0.24	0.95
Ash (%dry matter)	4.93	5.04	0.06	0.45

^1^ Pooled standard error of the mean. ^2^ The results are expressed as optical density from ingested bacterial cells.

## Data Availability

The data that support the findings of this study are available on request from the corresponding author.

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
