# Peer review of "Effects of Butterfly Pea Extracts on Phagocytic Activity of Blood Polymorphonuclear Leukocytes and Muscular Lipid Peroxidation in Rabbits"

_animals, 2024, doi:10.3390/ani14060958_

Round 1

Reviewer 1 Report

Comments and Suggestions for Authors

The aim of this study was to evaluate the possible effects of Clitoria Ternatea crude extract supplementation on rabbits’ immunity and health status. In my opinion, the manuscript, although well written, highlights some basic problems . The productive context in which the research is being developed is not clear; it is not described how rabbits were raised, whether rabbit farming in Thailand is a significant livestock activity, much less whether rabbit meat is consumed in a significant way. Furthermore, the number of animals used is very low; this is a critical point of the research, mainly for the part relating to the carcass. It is not explained whether the use of Butterfly Pea can have an application in the production chain (I have serious doubts). Perhaps the work could be set up as a study in which the rabbit represents an animal model for verifying the metabolic effects of the essence in the human health. In its current approach it is halfway between a productive and a physiological meaning which could confuse the reader.

Reviewer 2 Report

Comments and Suggestions for Authors

Manuscript ID: animals-2895401

Title: Effects of Butterfly Pea Extracts on Phagocytic Activity of Blood Polymorphonuclear Leukocytes and Muscular Lipid Peroxidation in Rabbits

Overall, it's a pretty well written manuscript. However, there are still many points that need improvement.

To improve the quality of the paper, update the reference list by adding 2020-2024 references, especially in the discussion section.

The study was well conducted but suffers from a small sample size, n=16. I am concerned that is a major limitation of this study, and as such is less informative.

Lines 62-66: The sentence began with this word "Recently", but at the end it ended with an old reference in 2011. Where is the modernity?

Line 94: This reference "Suwannamanee [21]" was "unpublished M.S. thesis " as you mentioned in the references list. Why has this reference not been published even though it has been since 2014?  It is preferable not to use it as long as it is not published.

I highly recommend adding the active components contained in the extract to this paper.

Why did you choose to use this level "0.5 g/kg body weight" in the study?

Line 162: "pH24"?

In Table 2: Please check the value of SEM of Yellowness " 0.36"

In Table 2: Please check the value of SEM of Ether extract " 0.24"

The discussion section needs strengthening. The authors should discuss the results clearly and should provide relevant information.

At the end of discussion, add a paragraph on describing the limitations of this work and practical applications.

Insert the correct format style for journals in the references in the text and references list.

Comments on the Quality of English Language

Moderate editing of English language required

Reviewer 3 Report

Comments and Suggestions for Authors

Manuscript: Effects of Butterfly Pea Extracts on Phagocytic Activity of Blood Polymorphonuclear Leukocytes and Muscular Lipid Peroxidation in Rabbits

Comments to the Authors

LN: Line Number

LN21: Please include the BW and SD details, age and the sex of rabbits in parenthesis.

LN22: Include the treatment group details please.

LN25: Apparent digestibility of what? Please mention.

LN32: Apparent digestibility of what?...Please mention.

LN101: Please mention the ethical approval details. Institute, approval number and the date.

LN104: Please mention the sex of the animals used.

LN104: What is the age of animals at start? Please mention.

LN105: Please mention the cage dimensions (L x W x H)

LN107: Any justification for not using replicates in your study? The validity of data based on 08 animals is questionable.

LN108: Why did you use a single level? Any justification? Please provide.

LN110: Please include the diet details (ME and CP) in parenthesis and also the form of the diets (Pellets?)

LN113: and ash by ignition to 550C will mislead the reader. This is important only when analyzing ash. Not the rest. So delete.

LN116: This has been done only for the control diet. Why not for CTE?. Please explain.

LN121: This should be corrected as: Analyzed chemical composition.... This Table must be moved to the Results section. 

LN127: Please delete: for and replace with: until

LN129: Which nutrient digestibility values? Please mention in parenthesis.

LN130: keep a space: 10 mm.

LN145: How did you analyze the samples? In Duplicates or triplicates? Please mention.

LN159: What was the slaughtering method adopted? Please mention.

LN183: The results of chemical analysis of the diets and CTE should be discussed here.

LN203: Only the statistical significance attributes must be brought into discussion.

LN292: This should be corrected as Pooled Standard Error of Mean.

LN309: Only OM and EE instabilities were improved not all the nutrient digestibility values. Hence correct. 

Comments on the Quality of English Language

Needs minor editing.

Round 2

Reviewer 1 Report

Comments and Suggestions for Authors

Dear Authors, in my opinion all the observations have been adequately received and the relative corrections made. So, in my opinion, the manuscript is now suitable for publication on Animals

Reviewer 3 Report

Comments and Suggestions for Authors

Manuscript: Effects of Butterfly Pea Extracts on Phagocytic Activity of Blood Polymorphonuclear Leukocytes and Muscular Lipid Peroxidation in Rabbits

Comments to the Authors

LN: Line Number

LN24: Sixteen, 35-day-old...treatment group. Keep a full-stop here. Please revise the next sentence as: The treatment group was orally administered with crude extracts .......to slaughter. Please delete the word: administration o (LN24)

LN24: administration: Please delete.

LN27: Propose to use: of instead of on...

LN34: Propose to use: of instead of on...

LN103-104: Please shift this sentence up and place immediately after Materials and Methods section.

LN111: Please revise as: (L x W x H: 45 cm x 45 cm x 45 cm)

LN122: suggest to revise as…were checked once in every two weeks.......

LN212: 213, 214: Use p, instead of P.

LN354: Table 2: Keep only SEM here and describe the full term in footnotes.

LN373: Please revise as: ....did improve the apparent digestibility of organic matter and ether extract.

Comments on the Quality of English Language

Need minor editing, specially the revised sections. 
